# Understanding the Fine-Grained Knowledge Capabilities of Vision-Language Models

## Abstract

Vision-language models (VLMs) have made substantial progress across a wide range of visual question answering benchmarks, spanning visual reasoning, document understanding, and multimodal dialogue. These improvements are evident in a wide range of VLMs built on a variety of base models, alignment architectures, and training data. However, recent works show that these models trail behind in traditional image classification benchmarks, which test fine-grained visual knowledge. We test a large number of recent VLMs on fine-grained classification benchmarks and identify potential factors in the disconnect between fine-grained knowledge and other vision benchmarks. Through a series of ablation experiments, we find that using a better LLM improves all benchmark scores equally, while a better vision encoder disproportionately improves fine-grained classification performance. Furthermore, we find that the pretraining stage is also vital to fine-grained performance, particularly when the language model weights are unfrozen during pretraining. These insights pave the way for enhancing fine-grained visual understanding and vision-centric capabilities in VLMs.

## 1 Introduction

Recent advances in vision-language models (VLMs), which integrate vision encoders with large language models (LLMs), have demonstrated increasingly sophisticated capabilities in interpreting and reasoning about visual content (Alayrac et al., 2022; Team et al., 2023; OpenAI, 2023; Liu et al., 2023; Wang et al., 2024b). These models can perform complex tasks such as open-ended visual question answering, document understanding, and multimodal dialogue (Goyal et al., 2017; Yue et al., 2024; Mathew et al., 2021; OpenAI, 2023). Despite these advances, a crucial question remains: How well do VLMs perform on fine-grained visual perception tasks, and how can we improve their performance?

Fine-grained visual recognition—the ability to distinguish between visually similar categories—requires models to focus on subtle distinguishing features while maintaining robustness to intra-class variations in pose, lighting, and background. While traditional vision encoders like CLIP (Radford et al., 2021) and DINO (Caron et al., 2021) have demonstrated strong performance on fine-grained classification benchmarks, the performance of newer VLMs has received less attention.

Understanding the fine-grained knowledge capabilities of VLMs is essential because perception serves as the foundation for advanced capabilities such as understanding and reasoning. Real-world applications of VLMs often involve fine-grained, long-tailed distributions where distinguishing between closely related categories is crucial. For instance, if a model fails to identify a mushroom species correctly, it cannot determine whether it is poisonous based on textual knowledge or whether it is safe for consumption. Moreover, insights into the factors influencing fine-grained knowledge acquisition could inform more effective architectures and training strategies for future VLMs.

In this paper, we first present a comprehensive evaluation of fine-grained visual knowledge in state-of-the-art VLMs, comparing their performance on fine-grained classification benchmarks with their capabilities on general VLM tasks. Specifically, we evaluate 15 VLMs, such as LLaVA (Liu et al., 2023; 2024a), Phi (Abdin et al., 2024), Qwen2-VL (Wang et al., 2024b), Molmo (Deitke et al., 2024), on four fine-grained classification benchmarks, including ImageNet (Deng et al., 2009), Flowers (Nilsback & Zisserman, 2008), Pets (Parkhi et al., 2012), and Food (Bossard et al., 2014), alongside aggregated results from eight general VLM benchmarks such as MMMU (Yue et al.,

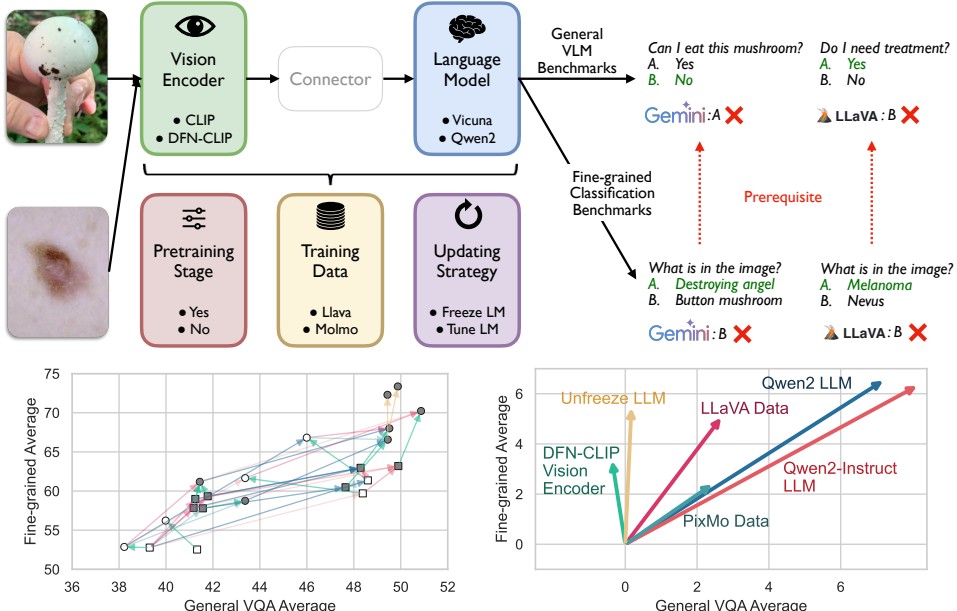

Figure 1: **Overview.** *(Top)* We investigate the fine-grained classification capabilities of vision-language models (VLMs), a crucial yet often overlooked aspect that underpins higher-level understanding and reasoning. *(Bottom)* Through 22 systematic ablation experiments on key model components and training strategies (left), we identify the factors that drive fine-grained classification performance in VLMs (right).

2024), MathVista (Lu et al., 2024), and MMVet (Yu et al., 2024). Our analysis reveals significant variance in fine-grained classification performance among models with comparable general abilities, suggesting that fine-grained recognition represents a distinct axis of visual intelligence not captured by existing VLM benchmarks. Furthermore, we observe a persistent gap between VLMs and their CLIP-based vision encoders, mirroring previous findings (Zhang et al., 2024; Geigle et al., 2024).

To better understand what contributes to fine-grained classification performance in VLMs, we conduct a systematic study examining the role of key model components and training paradigms. Specifically, we investigate whether VLM components, such as language models and vision encoders, impact classification performance and whether training strategies, such as pretraining or variations in data qualities, influence results. Through 22 carefully designed ablation experiments, we identify several factors that significantly affect fine-grained knowledge acquisition in VLMs:

- **Language model (§4.2):** Improvements in the base language model enhance performance uniformly across both fine-grained classification and general VLM benchmarks.
- **Vision encoder (§4.2):** A stronger vision encoder improves fine-grained classification performance but has a limited impact on general VLM benchmarks, particularly in models trained in a two-stage fashion.
- **Pretraining stage (§4.3):** Large-scale pretraining on image-captioning datasets significantly boosts fine-grained classification performance but has a lesser effect on general VLM benchmarks.
- **Weight updating method (§4.3):** Fine-grained classification performance improves when both the connector between the vision encoder and LLM, as well as the LLM itself, are trained during pretraining, compared to training only the connector.
- **Data quality (§4.3):** The quality of pretraining and instruction-tuning data has a limited impact on fine-grained classification, particularly when language model weights are frozen during pretraining.

In summary, our study examines fine-grained visual perception in VLMs—an essential yet overlooked aspect of current VLM benchmark evaluations. Through systematic ablations, we identify key factors influencing fine-grained knowledge acquisition in VLMs. These findings offer valuable insights for developing vision-centric VLMs with enhanced visual understanding, improving their effectiveness in real-world long-tail tasks that demand fine-grained recognition.

## 2 RELATED WORK

**Vision-language models and their evaluation.** Vision-language models (VLMs) are a class of models that integrate visual inputs with language models, typically following the LLaVA-based architecture (Liu et al., 2023) of three core components: a vision encoder, a language model, and an MLP connector bridging the two. These models have achieved strong performance across various multimodal tasks, including visual question answering (VQA), document understanding, and general reasoning (Liu et al., 2023; Dai et al., 2023; Awadalla et al., 2023; Wang et al., 2024b). To evaluate VLMs, numerous benchmarks have been developed, such as MMMU (Yue et al., 2024), MathVista (Wang et al., 2024a), and DocVQA (Mathew et al., 2021). However, these benchmarks primarily assess reasoning and language understanding given visual inputs, while overlooking core vision-centric capabilities such as object recognition and fine-grained classification. To bridge this gap, in this work we systematically evaluate and analyze VLMs on fine-grained classification, an essential yet underexplored aspect of visual intelligence.

**Fine-grained visual classification.** Fine-grained classification is a well-established computer vision task that aims to distinguish between visually similar subcategories within broader object categories, such as bird species (Wah et al., 2011), flowers (Nilsback & Zisserman, 2008), and pet breeds (Parkhi et al., 2012). These benchmarks have played a key role in training and evaluating vision encoders, like ResNet (He et al., 2016), CLIP (Radford et al., 2021) and DINO (Caron et al., 2021). Despite their importance in vision research, recent VLMs have largely overlooked these datasets in their evaluation protocols in favor of visual question-answering. Since fine-grained classification is critical for real-world applications that involve long-tail distributions and require precise recognition (e.g., medical diagnosis, food safety, and species identification), evaluating VLMs on such benchmarks is essential for understanding their practical usability from a vision-centric perspective, beyond language-centric reasoning tasks.

**Evaluation and ablation of VLMs for fine-grained classification.** While recent studies have explored VLM evaluation on fine-grained classification (Zhang et al., 2024; Geigle et al., 2024), they have been limited in scope, often focusing on just a few models or datasets. We extend these efforts by evaluating a broader range of VLMs across multiple fine-grained classification benchmarks in a unified multiple-choice format. Additionally, we go beyond prior work with a comprehensive ablation analysis to identify key factors that impact fine-grained classification performance.

Several recent works have investigated the design space of VLMs, analyzing aspects such as model architectures (McKinzie et al., 2024), training strategies (Karamcheti et al., 2024), and the role of multimodal datasets (Gadre et al., 2023; Udandarao et al., 2024; Deitke et al., 2024). However, these studies primarily focus on general benchmarks that are more language and reasoning-focused. In this work, we investigate how key design choices—including base model selection, pretraining methods, and the composition of pretraining and instruction-tuning data—affect fine-grained classification, providing novel insights into optimizing VLMs for vision-centric tasks.

## 3 EVALUATION: FINE-GRAINED CLASSIFICATION

In this section, we discuss the motivation for evaluating fine-grained classification, describe our benchmarks and evaluation setup, and present our key observations from testing 15 existing VLMs.

### 3.1 MOTIVATION

Fine-grained classification is a fundamental task in computer vision that focuses on distinguishing visually similar subcategories within a broader category (e.g., different bird species or car models) (Wei et al., 2021). Many evaluation benchmarks, such as ImageNet (Deng et al., 2009) and Flowers (Nilsback & Zisserman, 2008), have been developed and widely used as standard testbeds for training and evaluating vision encoders. With the advent of large-scale self-supervised and supervised learning, vision encoders such as CLIP (Radford et al., 2021) and DINO (Caron et al., 2021) achieved near- or superhuman performance on challenging fine-grained classification tasks.

However, these benchmarks have been overlooked in the evaluation of vision-language models (VLMs). While recent VLMs like LLaVA (Liu et al., 2023) have demonstrated strong capabilities in interpreting visual concepts, their evaluation is typically limited to visual question answering

(VQA), such as academic VQA benchmarks like MMMU (Yue et al., 2024) and MathVista (Lu et al., 2024), or document understanding tasks like DocVQA (Mathew et al., 2021) and TextVQA (Singh et al., 2019). These benchmarks emphasize reasoning and language processing but lack fine-grained object categories, thereby leaving core vision-centric capabilities largely unexamined.

Fine-grained classification is crucial for VLMs, as real-world applications often encounter long-tailed distributions that require precise visual knowledge between similar categories. For example, in daily life, if a blind person asks whether a mushroom is safe to eat, the model must first correctly identify the species before reasoning about its toxicity—misclassifying a deadly destroying angel as an edible white button mushroom could have fatal consequences. Similarly, in medical diagnosis, failing to distinguish between similar diseases could lead to incorrect treatment and serious health risks. In self-driving cars, confusing a "stop" sign with a "do not enter" sign—both red with similar shapes—could result in dangerous navigation errors. Therefore, it is critical for VLMs to excel at fine-grained classification to ensure safety and reliability in real-world scenarios.

## 3.2 BENCHMARKS

We evaluate the fine-grained visual knowledge of VLMs using four well-established object recognition benchmarks, each focusing on different domains of fine-grained classification: **ImageNet-1K** (Deng et al., 2009), **Oxford Flowers-102** (Nilsback & Zisserman, 2008), **Oxford-IIIT Pet-37** (Parkhi et al., 2012), and **Food-101** (Bossard et al., 2014).

These benchmarks contain 37 to 1,000 distinct classes, making direct evaluation of VLMs in a multiple-choice format—commonly used in VLM training and evaluation—challenging. To address this, we adopt the methodology of Geigle et al. (2024) to convert these datasets into 5-way multiple-choice questions. Specifically, for each image, we use an OpenCLIP ViT-L/14 model (Schuhmann et al., 2022) to select the four hardest negative choices based on image-label cosine similarity and combine them with the ground-truth label to form the multiple-choice answers. We choose Open-CLIP model to avoid biasing the difficulty against VLMs which rely on OpenAI-CLIP (Radford et al., 2021), SigLIP (Zhai et al., 2023), or DFN-CLIP (Fang et al., 2023) (which are all trained on different datasets than the OpenCLIP model). This conversion to multiple choice allows us to test VLMs in their native format while preserving much of the difficulty. For instance, CLIP ViT-L/14-336px (Radford et al., 2021) obtains 75.2% accuracy on the 102-class Flowers-102, which only increases to 78.3% accuracy on our 5-way multiple choice version.

## 3.3 TESTING SETUP

We evaluate 15 recent VLMs in the 7B–13B parameter range on the converted 5-choice fine-grained classification benchmarks (Liu et al., 2023; 2024a; Chen et al., 2024a; Abdin et al., 2024; Deitke et al., 2024; Laurençon et al., 2024; Hong et al., 2024; Chen et al., 2024c; Wang et al., 2024b). For each VLM, we use the respective default prompt format from VLMEvalKit (Duan et al., 2024) and measure accuracy based on the exact match with the multiple choice options. For baseline comparison, we evaluate various CLIP models (Radford et al., 2021; Schuhmann et al., 2022; Fang et al., 2023) in the standard zero-shot setup, where both images and class labels ("A photo of a [class]") are encoded using CLIP's image and text encoders. The top-1 prediction is determined by computing the cosine similarity between image and text embeddings.

To assess whether fine-grained classification capabilities are captured by existing VLM evaluation benchmarks, we compare VLM performance on fine-grained classification benchmarks to their average score on eight general VQA benchmarks, such as MMMU (Yue et al., 2024) and MathVista (Lu et al., 2024), from VLMEvalKit.

## 3.4 RESULTS

Figure 2 shows the relationship between fine-grained classification performance (averaged across our four benchmarks) and general VQA performance (averaged across eight existing VQA benchmarks) for all tested VLMs. We observe that models with similar performance on general VQA benchmarks exhibit substantial variation in fine-grained classification accuracy. For example, while CogVLM-Chat (Hong et al., 2024) and LLaVA-NeXT-Vicuna-13B (Liu et al., 2024a) both achieve nearly 48% average accuracy on general VQA benchmarks, their fine-grained classification accu-

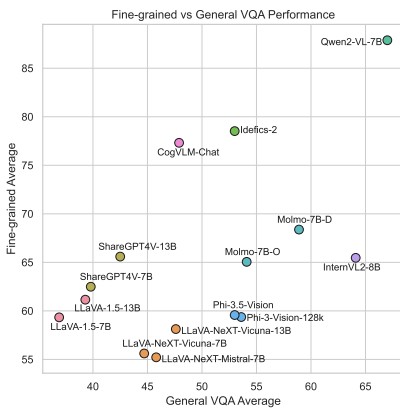

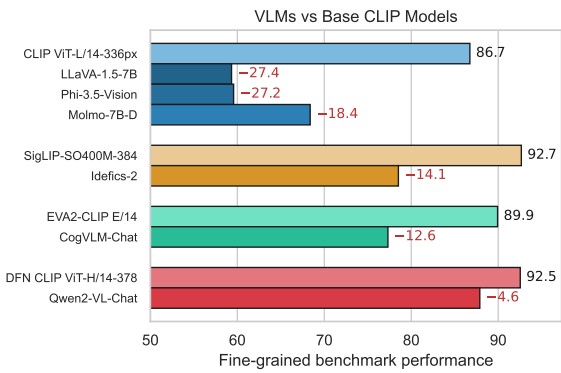

Figure 2: **Fine-grained classification compared to general VQA performance across VLMs.** Analysis of recent VLMs indicates that fine-grained classification represents a distinct aspect of visual capability that standard VQA benchmarks fail to measure.

Figure 3: **Comparison of VLMs with their corresponding CLIP vision encoders in fine-grained classification.** While Qwen2-VL-Chat nearly matches the performance of its vision encoder DFN-CLIP, all other VLMs fall significantly behind. This highlights that VLMs have considerable room for improvement in fine-grained classification tasks.

racy differs by 19pp (percentage points)—77.3% for CogVLM-Chat and 58.1% for LLaVA-NeXT-Vicuna-13B. This large discrepancy indicates that fine-grained classification capabilities are not well captured by existing VLM evaluation benchmarks, highlighting fine-grained classification as a distinct dimension of visual intelligence.

> ***Finding 1:*** Fine-grained classification represents a distinct aspect of visual capability that current VLM benchmarks fail to adequately measure.

An important aspect of current VLMs is that they each build on an existing pretrained vision encoder. This means that with the multiple-choice format, we can directly compare performance between VLMs and CLIP models on traditional classification benchmarks. Figure 3 compares the fine-grained classification performance of selected VLMs with their corresponding CLIP model baselines, revealing a substantial gap between them. For instance, Molmo (Deitke et al., 2024), a very recent model which excels on different VQA benchmarks obtains only a 68.4% average accuracy in our multiple choice evaluation setting, while its vision encoder CLIP ViT-L/14 (Radford et al., 2021) obtains 86.7%. Even the best tested model, Qwen2-VL-Chat (Wang et al., 2024b), falls 4.6 points behind its own state-of-the-art DFN-CLIP (Fang et al., 2023) vision encoder. This performance gap underscores the need to analyze different design choices in VLMs to identify key factors that enhance fine-grained classification capabilities. In the next section, we conduct an ablation study to investigate these factors.

> ***Finding 2:*** A significant performance gap exists between VLMs and CLIP models in fine-grained classification tasks.

## 4 ABLATIONS: BASE MODELS AND TRAINING

### 4.1 EXPERIMENTAL SETUP

Building on our observational results in Section 3, we ablate key differences between models that may contribute to fine-grained classification performance, focusing primarily on base model selection and training methodology. We build on the LLaVA-1.5 training framework and alignment architecture (Liu et al., 2023). All models undergo one epoch of pretraining (when applicable) followed by one epoch of finetuning. We use LLaVA-1.5's default hyperparameters, including learning rate and batch size, across all experiments.

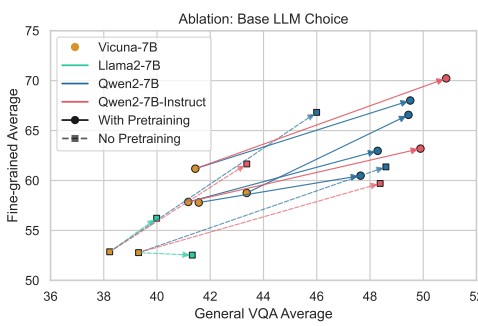 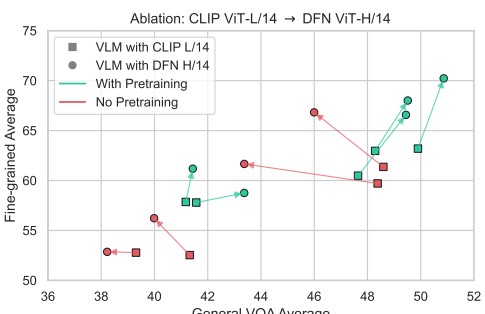

Figure 4: **Ablating base LLM, swapping out Vicuna-7B from LLaVA with other LLMs.** On average, switching from Vicuna to Qwen2-7B results in a +7.5pp increase in fine-grained performance and a +7.7pp improvement in general VQA performance.

Figure 5: **Ablating base vision encoder, swapping out CLIP ViT-L/14 from LLaVA for DFN-CLIP ViT-H/14.** We find that switching to DFN-CLIP improves fine-grained performance by +4.5pp and general performance by +1.2pp, given enough pretraining.

Fine-grained performance is evaluated as the average score across the four classification benchmarks described in Section 3.2. For general VQA performance, we report the average score on three widely-used multiple-choice benchmarks: MMMU (Yue et al., 2024), MMBench (Liu et al., 2024b), and MMStar (Chen et al., 2024b). These benchmarks were selected for their domain coverage and widespread adoption in VLM evaluation.

## 4.2 BASE MODELS

Our analysis in Section 3 indicates that recent models typically incorporate newer base LLMs, though most continue to use the CLIP ViT-L/14 vision encoder from Radford et al. (2021). We investigate how these base model choices specifically impact fine-grained classification performance.

### 4.2.1 LLM CHOICE

**Setup.** Using LLaVA-1.5-7B (Liu et al., 2023) as our baseline, we ablate the LLM component across different vision encoders and pretraining configurations. We compare the original Vicuna-7B-1.5 (Chiang et al., 2023) (used in LLaVA) against Qwen2-7B (Yang et al., 2024) (used in Qwen2-VL-7B (Wang et al., 2024b) and Molmo-7B-D (Deitke et al., 2024)), as well as Llama-2-7B (Touvron et al., 2023) (Vicuna's base model) and Qwen2-7B-Instruct (the instruction-tuned variant of Qwen2-7B).

**Findings.** As shown in Figure 4, replacing Vicuna-7B with Qwen2-7B substantially improves performance across all configurations, with average gains of +7.5pp (percentage points) on fine-grained benchmarks and +7.4pp on general VQA benchmarks. The instruction-tuned Qwen2-7B-Instruct shows similar improvements (+7.5pp and +8.1pp, respectively). Llama2-7B, the non-instruction-tuned base of Vicuna, yields modest improvements, though the downstream implications of using a model without instruction tuning remain unclear. Notably, all our trained models—including the lower-performing Vicuna-based VLMs—consistently produce multiple-choice responses in the correct format. This suggests that performance differences in fine-grained classification stem from improved knowledge about the presented options rather than better format adherence.

> ***Takeaway 1:*** Stronger language models consistently improve performance across both fine-grained classification and general VQA benchmarks.

### 4.2.2 VISION ENCODER CHOICE

**Setup.** Most evaluated VLMs, including the LLaVA series and Molmo, use OpenAI-CLIP ViT-L/14-336px (Radford et al., 2021) as their vision encoder. However, Qwen2-VL (Wang et al., 2024b) uses DFN-CLIP ViT-H/14-378px (Fang et al., 2023), which demonstrates superior zero-shot perfor-

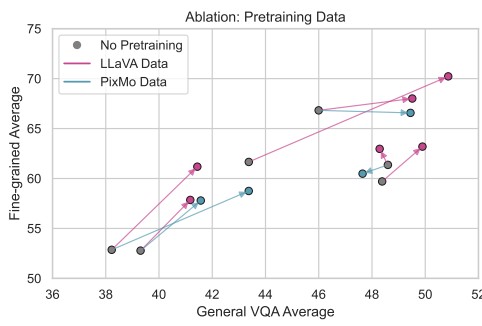 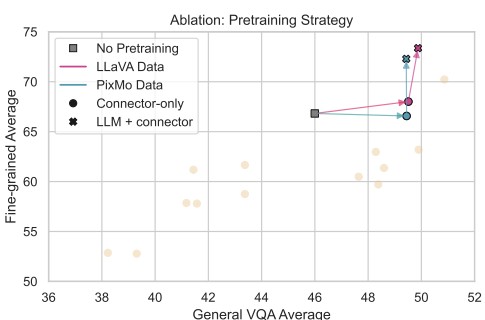

Figure 6: **Ablating pretraining data.** Adding either LLaVA (CC-3M) or Molmo (PixMo-Cap) data gives similar gains. LLaVA data increases fine-grained and general scores by +4.0pp and +2.1pp respectively, while PixMo data increases them by +2.4pp and +2.5pp.

Figure 7: **Ablating training method, comparing connector-only pretraining with LLM+connector tuning.** Unfreezing the LLM during pretraining yields a +5.5pp improvement in fine-grained performance without adversely affecting general VQA scores.

mance on object recognition tasks like ImageNet. We evaluate both vision encoders across various LLM choices and pretraining configurations.

**Findings.** Figure 5 illustrates that the impact of substituting CLIP L/14 with DFN-CLIP H/14 depends on whether connector pretraining is performed before finetuning. Without pretraining, DFN-CLIP degrades general VQA performance (-2.5pp) while modestly improving fine-grained benchmark scores (+2.8pp). However, on models that have undergone connector pretraining, general VQA performance remains stable or slightly improves (+1.2pp) while fine-grained performance increases substantially (+4.5pp). This indicates that enhanced vision encoders can significantly boost fine-grained classification performance in pretrained VLMs.

> *Takeaway 2:* Better vision encoders (like DFN-CLIP vs. CLIP) in VLMs improve fine-grained classification, but only when these encoders are properly integrated through pretraining before the finetuning stage.

## 4.3 TRAINING METHODS

Recent VLMs are typically trained in two stages (Liu et al., 2023; Wang et al., 2024b; Chen et al., 2024c). The first stage, pretraining, involves training on large-scale image-caption datasets using a captioning task to establish alignment between the vision encoder and LLM. The second stage, instruction tuning, refines the VLM with carefully curated multi-round, multi-modal instruction completion data, aligning it with human intent for real-world applications. Here, we explore the impact of these two training stages on the fine-grained capabilities of VLMs.

### 4.3.1 PRETRAINING

**Setup.** We investigate several critical questions about the pretraining process:

- Is pretraining necessary for LLaVA-architecture models when the base models are already pretrained on billion-scale data?
- How does connector-only pretraining compare with unfreezing the LLM?
- How important is the quality of pretraining data for downstream performance?

We experiment with three pretraining configurations: LLaVA pretraining data (Liu et al., 2023) (a subset of CC-3M with web-scraped captions), Molmo pretraining data (Deitke et al., 2024) (PixMo-Cap, with high-quality human-annotated detailed captions), and no pretraining. For most experiments, we pretrain only the connector following Liu et al. (2023). Additionally, we explore a strategy similar to Deitke et al. (2024), where we pretrain the connector for 20% of the steps before unfreezing the LLM for the remaining 80%. All configurations are followed by one epoch of finetuning on LLaVA finetuning data.

Table 1: **Summary of important ablation settings and comparison to existing models (bolded).** Upgrading the base LLM and vision encoder, as well as pretraining the connector and LLM on sufficient data, provide benefits to fine-grained performance, while instruction finetuning has a smaller effect. This is in contrast to general VQA benchmarks, where choice of LLM plays a larger part.

| Model/Ablation | Vision Encoder | LLM | Arch. | Pretraining Data | Finetuning Data | Fine-grained Classification | General VQA |
|---|---|---|---|---|---|---|---|
| **LLaVA-1.5-7B** | CLIP L/14 | Vicuna | LLaVA | LLaVA | LLaVA | 59.3 | 41.8 |
| No pretraining | CLIP L/14 | Vicuna | LLaVA | None | LLaVA | 52.8 | 39.3 |
| LLaVA reproduction | CLIP L/14 | Vicuna | LLaVA | LLaVA | LLaVA | 57.9 (+5.1) | 41.2 (+1.9) |
| Qwen2 LLM | CLIP L/14 | Qwen2 | LLaVA | LLaVA | LLaVA | 63.0 (+5.1) | 48.3 (+7.1) |
| DFN-CLIP encoder | DFN H/14 | Qwen2 | LLaVA | LLaVA | LLaVA | 68.0 (+5.0) | 49.5 (+1.2) |
| Unfreeze LLM | DFN H/14 | Qwen2 | LLaVA | LLaVA | LLaVA | 73.4 (+5.4) | 49.9 (+0.4) |
| Pretrain on PixMo | CLIP L/14 | Qwen2 | LLaVA | PixMo | LLaVA | 66.6 | 49.4 |
| **Molmo-7B-D** | CLIP L/14 | Qwen2 | Molmo | PixMo | Molmo | 68.4 | 58.0 |
| FT Qwen2-VL base on LLaVA | DFN H/14 | Qwen2 | Qwen2-VL | Qwen2-VL | LLaVA | 85.5 | 61.1 |
| **Qwen2-VL-7B** | DFN H/14 | Qwen2 | Qwen2-VL | Qwen2-VL | Qwen2-VL | 87.9 (+2.4) | 62.4 (+1.3) |

**Findings.** Figure 6 illustrates our pretraining ablation results. We observe that connector-only pretraining generally enhances VLM performance, with the exception of the Qwen2 and CLIP ViT-L/14 combination. The benefits are particularly substantial for Vicuna-based VLMs, which show an average improvement of +6.1pp in fine-grained classification and +3.1pp in general VQA performance.

> *Takeaway 3:* Large-scale pretraining on image captioning datasets substantially improves fine-grained classification performance but has a more modest effect on general VLM benchmarks.

Surprisingly, we find minimal differences between pretraining on low-quality web-scraped captions (LLaVA) versus highly detailed human annotations (PixMo). Compared to LLaVA data, PixMo data results in a -1.6pp change in fine-grained performance and a +0.4pp change in general VQA scores.

> *Takeaway 4:* Pretraining data quality has a limited impact on overall model performance.

We hypothesize that connector-only pretraining may prevent the model from fully leveraging higher-quality captions. To test this hypothesis, we experiment with pretraining both the LLM and connector, incorporating a 20% connector-only warmup phase. Figure 7 shows that this approach significantly enhances fine-grained classification performance (+5.5pp) without compromising general VQA scores. However, this improvement occurs with both LLaVA and PixMo pretraining data, suggesting that training on the lower-quality captions does not impair the LLM's language capabilities.

> *Takeaway 5:* Pretraining both LLM and connector substantially enhances fine-grained benchmark performance while maintaining general VQA scores.

### 4.3.2 FINETUNING

**Setup.** Departing from our previous setup, we begin with the pretrained Qwen2-VL-7B-Base model (Wang et al., 2024b) (from which Qwen2-VL-7B-Chat is trained) and finetune on LLaVA-Instruct data (Liu et al., 2023) following our earlier experimental protocol. We compare these results with Qwen2-VL-7B-Chat, which employs 2M in-house examples for instruction finetuning (Wang et al., 2024b). This comparison helps isolate the effects of different finetuning datasets.

**Findings.** Our results indicate that finetuning solely on LLaVA finetuning data slightly reduces both fine-grained and general benchmark performance. Specifically, fine-grained scores decrease by 2.4pp (from 87.9% to 85.5%), while general benchmark scores drop by 1.3pp (from 62.4% to 61.1%). As illustrated in Figure 8, this represents the smallest contribution to fine-grained performance across all our ablations, suggesting that instruction finetuning plays a less critical role in fine-grained knowledge acquisition compared to base model selection and pretraining strategy.

> *Takeaway 6:* The instruction finetuning stage has comparatively less impact on fine-grained classification performance than other factors.

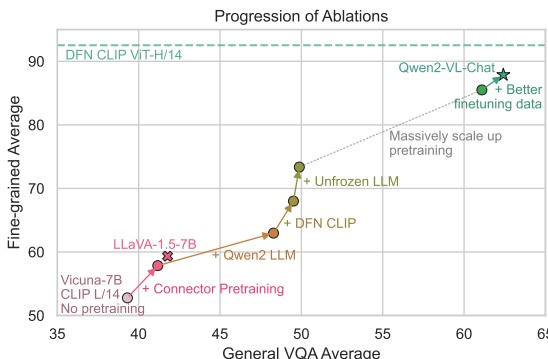

Figure 8: **Progression of combined ablation effects.** Starting with Vicuna-7B and CLIP ViT-L/14, we progressively modify the base LLM, vision encoder, and training settings to increase fine-grained classification accuracy from 52.8% to 73.4%, accounting for more than half the gap to Qwen2-VL-Chat's 87.9%. We attribute the remaining performance difference primarily to Qwen2-VL's extensive pretraining data of 1.4T tokens.

## 4.4 SUMMARY OF ABLATIONS

Figure 8 and Table 1 synthesize our ablation studies, showing how each component contributes to closing the performance gap between the weakest and strongest VLMs on both fine-grained and general VQA tasks. Beginning with the baseline LLaVA architecture (Liu et al., 2023) (Vicuna-7B (Chiang et al., 2023) and CLIP ViT-L/14 (Radford et al., 2021)), we systematically modified various components to increase fine-grained performance from 52.8% to 73.4%, and general VQA performance from 39.3% to 49.9%. This analysis reveals that switching to DFN-CLIP (Fang et al., 2023) and unfreezing the LLM during pretraining disproportionately enhances fine-grained capabilities compared to general VLM abilities, whereas switching the LLM to Qwen2-7B (Yang et al., 2024) accounts for most of the increase in general VQA performance.

Despite these substantial improvements, a notable 12-point gap in fine-grained classification performance remains unexplained by our ablations (Table 1). We consider two potential factors to account for this discrepancy: architectural differences and pretraining data scale.

While architectural changes could contribute to performance differences, our observational results (Section 3.4) and training ablations (Section 4.3) strongly suggest that pretraining data scale is the dominant factor. Our experiments with LLaVA (Liu et al., 2023) and PixMo (Deitke et al., 2024) data involved relatively small datasets—each comprising fewer than 1M images and captions, or approximately 200M and 400M tokens, respectively. In contrast, Wang et al. (2024b) report pretraining Qwen2-VL on an extensive 1.4T tokens—orders of magnitude more than in our experiments. This substantial disparity in pretraining data scale likely accounts for the remaining performance gap, highlighting the critical role of extensive pretraining in developing VLMs with superior fine-grained classification capabilities.

## 5 LIMITATIONS & CONCLUSION

Our study has some practical limitations that open up opportunities for future research. Due to computational constraints, we could only compare training on <1M data points rather than the billion (B) scale training utilized by some newer VLMs, leaving open the question of how large-scale training might impact our observations. Additionally, newer work suggest different pretraining strategies, which might interact differently with fine-grained visual understanding.

In this work, we systematically evaluate state-of-the-art vision-language models (VLMs) on fine-grained classification benchmarks, highlighting fine-grained visual classification as a crucial yet underexplored dimension of VLMs. Through an in-depth analysis of key model components and training paradigms, we provide insights into strategies for improving fine-grained classification and enhancing vision-centric capabilities, ultimately strengthening the applicability of VLMs in real-world scenarios that demand precise visual understanding.

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

| Vision Encoder | LLM | Arch | PT Data | FT Data | Updating | Classification | VLM |
|---|---|---|---|---|---|---|---|
| CLIP ViT-L/14@336 | Vicuna-7B | LLaVA | LLaVA | LLaVA | Connector | 59.3 | 41.8 |
| CLIP ViT-L/14@336 | Vicuna-7B | LLaVA | LLaVA | LLaVA | Connector | 59.0 | 41.2 |
| CLIP ViT-L/14@336 | Vicuna-7B | LLaVA | None | LLaVA | Connector | 52.8 | 39.3 |
| CLIP ViT-L/14@336 | Llama2-7B | LLaVA | None | LLaVA | Connector | 52.5 | 41.3 |
| DFN ViT-H/14@378 | Llama2-7B | LLaVA | None | LLaVA | Connector | 56.2 | 40.0 |
| DFN ViT-H/14@378 | Vicuna-7B | LLaVA | None | LLaVA | Connector | 52.9 | 38.2 |
| CLIP ViT-L/14@336 | Qwen2-7B | LLaVA | None | LLaVA | Connector | 61.4 | 48.6 |
| DFN ViT-H/14@378 | Qwen2-7B | LLaVA | None | LLaVA | Connector | 66.8 | 46.0 |
| CLIP ViT-L/14@336 | Qwen2-7B-Instruct | LLaVA | None | LLaVA | Connector | 59.7 | 48.4 |
| DFN ViT-H/14@378 | Qwen2-7B-Instruct | LLaVA | None | LLaVA | Connector | 61.7 | 43.4 |
| DFN ViT-H/14@378 | Qwen2-7B | Qwen2-VL | Qwen2-VL | LLaVA | Full | 85.5 | 61.1 |
| CLIP ViT-L/14@336 | Vicuna-7B | LLaVA | LLaVA | LLaVA | Connector | 57.9 | 41.2 |
| DFN ViT-H/14@378 | Vicuna-7B | LLaVA | LLaVA | LLaVA | Connector | 61.2 | 41.4 |
| CLIP ViT-L/14@336 | Qwen2-7B | LLaVA | LLaVA | LLaVA | Connector | 63.0 | 48.3 |
| DFN ViT-H/14@378 | Qwen2-7B | LLaVA | LLaVA | LLaVA | Connector | 68.0 | 49.5 |
| CLIP ViT-L/14@336 | Qwen2-7B-Instruct | LLaVA | LLaVA | LLaVA | Connector | 63.2 | 49.9 |
| DFN ViT-H/14@378 | Qwen2-7B-Instruct | LLaVA | LLaVA | LLaVA | Connector | 70.2 | 50.9 |
| DFN ViT-H/14@378 | Qwen2-7B | Qwen2-VL | Qwen2-VL | Qwen2-VL | Full | 87.9 | 62.4 |
| CLIP ViT-L/14@336 | Vicuna-7B | LLaVA | PixMo | LLaVA | Connector | 57.8 | 41.6 |
| DFN ViT-H/14@378 | Vicuna-7B | LLaVA | PixMo | LLaVA | Connector | 58.8 | 43.4 |
| CLIP ViT-L/14@336 | Qwen2-7B | LLaVA | PixMo | LLaVA | Connector | 60.5 | 47.6 |
| DFN ViT-H/14@378 | Qwen2-7B | LLaVA | PixMo | LLaVA | Connector | 66.6 | 49.4 |
| DFN ViT-H/14@378 | Qwen2-7B | LLaVA | LLaVA | LLaVA | Full | 73.4 | 49.9 |
| DFN ViT-H/14@378 | Qwen2-7B | LLaVA | PixMo | LLaVA | Full | 72.3 | 49.4 |

Table 2: **Detailed results of model ablations.**

| Vision Encoder | LLM | Arch | PT Data | FT Data | Updating | MMBench | MMMU | MMStar |
|---|---|---|---|---|---|---|---|---|
| CLIP ViT-L/14@336 | Vicuna-7B | LLaVA | LLaVA | LLaVA | Connector | 60.4 | 32.2 | 32.7 |
| CLIP ViT-L/14@336 | Vicuna-7B | LLaVA | LLaVA | LLaVA | Connector | 58.6 | 32.9 | 32.3 |
| CLIP ViT-L/14@336 | Vicuna-7B | LLaVA | None | LLaVA | Connector | 54.4 | 32.8 | 30.7 |
| CLIP ViT-L/14@336 | Llama2-7B | LLaVA | None | LLaVA | Connector | 58.0 | 34.3 | 31.7 |
| DFN ViT-H/14@378 | Llama2-7B | LLaVA | None | LLaVA | Connector | 53.9 | 34.6 | 31.5 |
| DFN ViT-H/14@378 | Vicuna-7B | LLaVA | None | LLaVA | Connector | 51.3 | 33.2 | 30.1 |
| CLIP ViT-L/14@336 | Qwen2-7B | LLaVA | None | LLaVA | Connector | 63.1 | 41.8 | 40.9 |
| DFN ViT-H/14@378 | Qwen2-7B | LLaVA | None | LLaVA | Connector | 61.3 | 36.2 | 40.5 |
| CLIP ViT-L/14@336 | Qwen2-7B-Instruct | LLaVA | None | LLaVA | Connector | 63.2 | 41.4 | 40.5 |
| DFN ViT-H/14@378 | Qwen2-7B-Instruct | LLaVA | None | LLaVA | Connector | 55.7 | 38.3 | 36.1 |
| DFN ViT-H/14@378 | Qwen2-7B | Qwen2-VL | Qwen2-VL | LLaVA | Full | 78.5 | 49.0 | 55.8 |
| CLIP ViT-L/14@336 | Vicuna-7B | LLaVA | LLaVA | LLaVA | Connector | 60.4 | 29.9 | 33.3 |
| DFN ViT-H/14@378 | Vicuna-7B | LLaVA | LLaVA | LLaVA | Connector | 57.4 | 34.2 | 32.7 |
| CLIP ViT-L/14@336 | Qwen2-7B | LLaVA | LLaVA | LLaVA | Connector | 64.4 | 41.3 | 39.1 |
| DFN ViT-H/14@378 | Qwen2-7B | LLaVA | LLaVA | LLaVA | Connector | 64.9 | 41.3 | 42.3 |
| CLIP ViT-L/14@336 | Qwen2-7B-Instruct | LLaVA | LLaVA | LLaVA | Connector | 66.5 | 41.3 | 41.9 |
| DFN ViT-H/14@378 | Qwen2-7B-Instruct | LLaVA | LLaVA | LLaVA | Connector | 68.0 | 41.6 | 43.0 |
| DFN ViT-H/14@378 | Qwen2-7B | Qwen2-VL | Qwen2-VL | Qwen2-VL | Full | 78.9 | 49.9 | 58.4 |
| CLIP ViT-L/14@336 | Vicuna-7B | LLaVA | PixMo | LLaVA | Connector | 57.8 | 32.6 | 34.3 |
| DFN ViT-H/14@378 | Vicuna-7B | LLaVA | PixMo | LLaVA | Connector | 61.1 | 34.1 | 34.9 |
| CLIP ViT-L/14@336 | Qwen2-7B | LLaVA | PixMo | LLaVA | Connector | 64.9 | 40.0 | 38.0 |
| DFN ViT-H/14@378 | Qwen2-7B | LLaVA | PixMo | LLaVA | Connector | 63.0 | 41.0 | 44.3 |
| DFN ViT-H/14@378 | Qwen2-7B | LLaVA | LLaVA | LLaVA | Full | 66.0 | 42.0 | 41.6 |
| DFN ViT-H/14@378 | Qwen2-7B | LLaVA | PixMo | LLaVA | Full | 63.7 | 40.0 | 44.6 |

Table 3: **Detailed results of model ablations on general VQA benchmarks.**

# A  FULL RESULTS

We present the complete ablation results for aggregated performance, general VQA benchmarks, and fine-grained classification in Table 2, Table 3, and Table 4, respectively.

| Vision Encoder | LLM | Arch | PT Data | FT Data | Updating | ImageNet | Flowers | Pets | Food |
|---|---|---|---|---|---|---|---|---|---|
| CLIP ViT-L/14@336 | Vicuna-7B | LLaVA | LLaVA | LLaVA | Connector | 70.9 | 50.7 | 46.3 | 69.4 |
| CLIP ViT-L/14@336 | Vicuna-7B | LLaVA | LLaVA | LLaVA | Connector | 68.7 | 50.4 | 46.5 | 70.3 |
| CLIP ViT-L/14@336 | Vicuna-7B | LLaVA | None | LLaVA | Connector | 64.3 | 46.1 | 40.4 | 60.4 |
| CLIP ViT-L/14@336 | Llama2-7B | LLaVA | None | LLaVA | Connector | 64.4 | 44.2 | 40.7 | 60.8 |
| DFN ViT-H/14@378 | Llama2-7B | LLaVA | None | LLaVA | Connector | 65.4 | 50.5 | 46.2 | 62.8 |
| DFN ViT-H/14@378 | Vicuna-7B | LLaVA | None | LLaVA | Connector | 62.3 | 48.7 | 39.2 | 61.3 |
| CLIP ViT-L/14@336 | Qwen2-7B | LLaVA | None | LLaVA | Connector | 70.6 | 44.1 | 55.6 | 75.2 |
| DFN ViT-H/14@378 | Qwen2-7B | LLaVA | None | LLaVA | Connector | 74.1 | 50.8 | 62.3 | 80.1 |
| CLIP ViT-L/14@336 | Qwen2-7B-Instruct | LLaVA | None | LLaVA | Connector | 71.4 | 39.6 | 53.1 | 74.7 |
| DFN ViT-H/14@378 | Qwen2-7B-Instruct | LLaVA | None | LLaVA | Connector | 72.0 | 46.0 | 54.1 | 74.5 |
| DFN ViT-H/14@378 | Qwen2-7B | Qwen2-VL | Qwen2-VL | LLaVA | Full | 84.8 | 79.5 | 87.1 | 90.7 |
| CLIP ViT-L/14@336 | Vicuna-7B | LLaVA | LLaVA | LLaVA | Connector | 69.1 | 50.0 | 45.3 | 67.1 |
| DFN ViT-H/14@378 | Vicuna-7B | LLaVA | LLaVA | LLaVA | Connector | 72.7 | 53.2 | 48.5 | 70.4 |
| CLIP ViT-L/14@336 | Qwen2-7B | LLaVA | LLaVA | LLaVA | Connector | 71.5 | 46.9 | 56.6 | 76.9 |
| DFN ViT-H/14@378 | Qwen2-7B | LLaVA | LLaVA | LLaVA | Connector | 76.1 | 54.1 | 59.1 | 82.7 |
| CLIP ViT-L/14@336 | Qwen2-7B-Instruct | LLaVA | LLaVA | LLaVA | Connector | 74.5 | 45.8 | 55.1 | 77.4 |
| DFN ViT-H/14@378 | Qwen2-7B-Instruct | LLaVA | LLaVA | LLaVA | Connector | 78.5 | 52.3 | 65.4 | 84.7 |
| DFN ViT-H/14@378 | Qwen2-7B | Qwen2-VL | Qwen2-VL | Qwen2-VL | Full | 85.9 | 82.3 | 91.0 | 92.3 |
| CLIP ViT-L/14@336 | Vicuna-7B | LLaVA | PixMo | LLaVA | Connector | 69.2 | 47.4 | 46.7 | 67.8 |
| DFN ViT-H/14@378 | Vicuna-7B | LLaVA | PixMo | LLaVA | Connector | 70.8 | 50.9 | 44.8 | 68.5 |
| CLIP ViT-L/14@336 | Qwen2-7B | LLaVA | PixMo | LLaVA | Connector | 70.8 | 43.7 | 53.9 | 73.6 |
| DFN ViT-H/14@378 | Qwen2-7B | LLaVA | PixMo | LLaVA | Connector | 76.1 | 50.3 | 57.8 | 82.1 |
| DFN ViT-H/14@378 | Qwen2-7B | LLaVA | LLaVA | LLaVA | Full | 79.4 | 60.7 | 67.3 | 86.0 |
| DFN ViT-H/14@378 | Qwen2-7B | LLaVA | PixMo | LLaVA | Full | 80.0 | 56.4 | 65.4 | 87.4 |

Table 4: **Detailed results of model ablations on fine-grained classification benchmarks.**

# B EVALUATION

## B.1 FINE-GRAINED CLASSIFICATION

As covered in Section 3.2, we evaluate VLMs and base vision encoders on four different traditional classification benchmarks:

**ImageNet-1K** (Deng et al., 2009): A foundational image classification dataset covering a broad range of supercategories, each containing multiple fine-grained subcategories, based on the WordNet hierarchy. The ImageNet test set contains 50,000 images with classification across 1,000 classes.

However, because of the nature of the ImageNet/WordNet class hierarchy, there are many classes which are too visually distinct from other classes to create hard multiple choice examples for (for instance, there are only three classes under the "person" category). Due to this drawback, in addition to the excessive size of the dataset, we employ human annotated incorrect model predictions from Recht et al. (2019) to select images for which hard negative answer choices can be constructed. Starting with 19,056 annotated examples, we narrow down the pool to 5,457 images consisting of 928 different classes out of the original 1,000. Furthermore, since this test example curation leaves the remaining classes unevenly balanced, we report the mean-per-class accuracy for ImageNet.

**Oxford Flowers-102** (Nilsback & Zisserman, 2008): A dataset of 102 flower species characterized by high intra-class variation and inter-class similarity. The Flowers dataset consists of 6,149 test examples.

**Oxford-IIIT Pet-37** (Parkhi et al., 2012): A collection of 37 pet categories with challenging variations in pose, lighting, and occlusion. The Pets dataset consists of 3,669 test examples.

**Food-101** (Bossard et al., 2014): A dataset comprising 101 food categories, exhibiting substantial visual diversity within each class. The Food dataset consists of 25,250 test examples.

**Question generation.** For all fine-grained classification test sets except the ImageNet dataset (described above), we generate hard negatives by performing zero-shot classification on the test examples using an OpenCLIP ViT-L/14 trained on LAION (Schuhmann et al., 2022), following Geigle et al. (2024). We take the top 4 predictions that are incorrect, and shuffle them with the correct label to produce the list of answer choices.

**Prompt formatting.** In our multiple choice VQA format, the question is always of the form "What type of object is in this photo?" The answer choices are formatted as "A. Option 1\n B. Option 2\n C. Option 3\n D. Option 4\n E. Option 5". However, since different VLMs are trained with different multiple choice prompts and format, we use each model's corresponding format from VLMEvalKit (Duan et al., 2024), for instance: "Answer with the option's letter from the given choices directly."

For CLIP models, we use the standard zero-shot evaluation procedure, and thus there is no question format. The classnames passed to the text encoder are instead formatted in the template of "a photo of a {classname}".

### B.2 GENERAL VQA

In our initial observational testing from Section 3, we rely directly on the numbers reported on the OpenVLM Leaderboard (Duan et al., 2024). The default leaderboard reports the average score across eight diverse VLM benchmarks: MMBench (Liu et al., 2024b), MMStar (Chen et al., 2024b), MMMU (Val) (Yue et al., 2024), MathVista (Lu et al., 2024), OCRBench (Liu et al., 2024c), AI2D (Kembhavi et al., 2016), HallusionBench (Guan et al., 2024), and MMVet (Yu et al., 2024). This reported average score is used only in Figure 2 in our paper.

For all further evaluations, both on existing models and our own models trained in Section 4, we narrow this down to a subset of multiple-choice VLM benchmarks for a more direct comparison of performance. Specifically, we evaluate VLMs on three benchmarks:

**MMBench** (Liu et al., 2024b): A dataset of 2,948 examples testing a wide range of VLM capabilities split between perception and reasoning. While the majority of the examples deal with coarse-grained perception and visual reasoning tasks, some examples test "fine-grained perception", though the categories (such as action recognition, attribute recognition, and OCR) are in a different domain from the fine-grained classification datasets we consider.

**MMMU (Val)** (Yue et al., 2024): A large-scale multimodal dataset covering a wide distribution of domains and applications. Though the test set has 10,500 examples, most works cite model performance on the validation split, which contains 900 questions. MMMU tests general knowledge tied to visual perception across art, science, engineering, medicine, business, and humanities.

**MMStar** (Chen et al., 2024b): A dataset comprised of 1,500 questions selected by human annotators. The examples are chosen to cover a diverse set of tasks, both across domain and low-level model capabilities.

**Evaluation procedure.** For the general VQA benchmarks, we defer entirely to the VLMEvalKit evaluation code (Duan et al., 2024). All existing models have a pre-defined evaluation procedure, and for our own trained models, we apply the LLaVA evaluation code, since we exclusively finetune on LLaVA-Instruct data.

## C TRAINING

We build on the LLaVA-1.5 training codebase (Liu et al., 2023). For the majority of experiments, we stick to the LLaVA architecture and training procedure.

**Pretraining.** This means one epoch of pretraining on the LLaVA pretraining data, obtained from CC-3M images and captions, or Molmo pretraining data (PixMo-Cap). For PixMo-Cap, we only use the "caption" field and not the raw audio transcript. During pretraining, we use a batch size of 256. In most of our pretraining runs, we tune only the randomly-initialized MLP connector with a learning rate of 1e-3. When ablating pretraining strategy, we first tune the connector only as warmup, for 20% of steps, then unfreeze the LLM and tune it as well, with a learning rate of 2e-5 for the remaining 80% of steps.

**Finetuning.** For finetuning, we always train for one epoch on the LLaVA-1.5 finetuning data, which consists of LLaVA-Instruct and VQA training data from various datasets to allow proper benchmarking. During finetuning, we maintain a learning rate of 2e-5 for the LLM, with a batch size of 128. The vision encoder remains frozen during both phases of training.

For the ablation experiment on the Qwen2-VL architecture, we start with Qwen2-VL-7B Base and train with the same method, unfreezing only the LLM.

**Compute.** We train all models using 4 nodes of 4 A100 (40GB) GPUs each. Training typically completes within 20 hours, with variation depending on the CLIP model size.

**LLaVA vs. PixMo data.** In our main text, we refer to the difference in caption quality between LLaVA pretraining data and PixMo pretraining data. LLaVA pretraining data is sourced from CC-3M, which consists of web-scraped captions, commonly known for low quality and relevance. PixMo-Cap is, on the other hand, collected from human annotators recording a 60–90 second audio description of an image, and then combined into a proper image caption (Deitke et al., 2024). A surface analysis of the two datasets shows that the LLaVA data averages 9.8 words per caption, whereas the PixMo data averages 169 words per caption.

*LLaVA caption example:* "front panel bracket cover for suzuki"

*PixMo caption example:* "The poster from the TV show "Law and Order" features a past cast ensemble set in an interrogation room. On the left is a Hispanic man, identified as Detective Nick Amaro, portrayed by Danny Pino. He has dark hair, olive skin, and is dressed in a gray jacket, blue shirt, and black tie. Standing beside him is a blonde woman with shoulder-length hair, who is identified as Detective Amanda Rollins, played by Kelli Giddish. She is ... The backdrop is an interrogation room, with a concrete wall and a portion of a two-way mirror visible, adding to the procedural drama's atmospheric setting."

