# OpenReview forum: "Understanding the Fine-Grained Knowledge Capabilities of Vision-Language Models"
_ICLR.cc/2026/Conference — Submitted to ICLR 2026_

### Official Review · Reviewer_ymmi · 2025-10-29

**Soundness:** 2
**Presentation:** 3
**Contribution:** 3
**Rating:** 6
**Confidence:** 3

**Summary:**

This paper examines whether modern vision–language models truly master fine-grained visual recognition. The authors build a unified five-choice evaluation from classic datasets, compare 15 models against CLIP baselines, and find weak alignment between general VQA scores and fine-grained ability. They show that strong encoders and caption-based pretraining, especially with the LLM unfrozen, are key to improving fine-grained performance and offer practical guidance for building better VLMs.

**Strengths:**

Overall, I think this work is interesting and has the potential to contribute to the community in the future. The Strengths are shown as follows:
1. Frames fine-grained recognition as a distinct evaluation axis for VLMs and operationalizes it with a unified five-choice protocol across classic datasets.
2. Provides a careful empirical study over 15 models with controlled ablations that tease apart the roles of the LLM, vision encoder, and pretraining strategy.
3. Presents methods and results clearly, with a consistent evaluation setup that makes cross-model comparisons straightforward.
4. Offers practically useful insights, especially the importance of caption-based pretraining and encoder choice, that are likely to influence how future VLMs are built and assessed.

**Weaknesses:**

1. Prompt sensitivity is under-quantified, since results rely on model-specific default prompts rather than a unified (or swept) prompting scheme, leaving unclear how much of the reported performance and ranking stems from prompt choices versus model capability.

2. Using CLIP both to mine hard negatives and to score CLIP baselines aligns dataset construction and evaluation with the CLIP embedding space, risking a systematic bias that can disadvantage generative VLMs and inflate CLIP-family comparability.

**Questions:**

Refer to the Weaknesses mentioned

---

### Official Review · Reviewer_ALgw · 2025-10-31

**Soundness:** 2
**Presentation:** 2
**Contribution:** 2
**Rating:** 2
**Confidence:** 5

**Summary:**

This paper evaluates state-of-the-art VLMs on fine-grained classification tasks alongside general VLM benchmarks. Through ablation studies of key model components and training paradigms of different VLMs, it provides some strategies for improving fine-grained classification and enhancing vision-centric capabilities.

**Strengths:**

1. It is well-motivated to investigate VLMs on traditional image classification benchmarks, which test fine-grained visual
knowledge of existing VLMs.
2. It ablates key differences between models that may contribute to fine-grained classification performance, providing some technical strategies for improving the performance.

**Weaknesses:**

1. The contribution of Chapter 3 is limited. The findings 1 and 2 have been discovered in [a]. Moreover, some typical fine-grained classification datasets, like CaltechUCSD Bird-200, Stanford Car-196, Stanford Dog-120, and FGVC-Aircraft are not included in the evaluation.
2. Some VLMs designed for FGVR are missing for comparison, like Finedefics [b] and DeepPerception [c].
3. The techinal depth is limited. Although it provides a series of ablation studies, further analysis on the potential reason (e.g., how it changes the fine-grained knowledge inside the model) is missing.
4. Some terminology is confusing. For example, are fine-grained visual perception, fine-grained visual recognition, and fine-grained knowledge capabilities refer to the same thing?

[a] Zheng et. al., Why are visually-grounded language models bad at image classification? NeurIPS 2024.

[b] He et. al., Analyzing and boosting the power of fine-grained visual recognition for multi-modal large language models, ICLR 2025.

[c] Ma et. al., Deepperception: Advancing r1-like cognitive visual perception in mllms for knowledge-intensive visual grounding, ArXiv 2025.

**Questions:**

1. The grammer of "fine-grained knowledge capabilities" seems incorrect.
1. ImageNet-1K is coarser than the other typical fine-grained classification datasets and commonly used for general recognition.
2. It is typical to use 4-way multiple-choice questions to evaluate the performance of fine-grained classification, why does the work use 5-way instead?
3. How about VLMs that are trained for three stages? How will the RLHF stage influence the performance?

---

### Official Review · Reviewer_eHLb · 2025-11-01

**Soundness:** 3
**Presentation:** 3
**Contribution:** 3
**Rating:** 6
**Confidence:** 3

**Summary:**

In addition to examining the general VQA performance of modern MLLMs, this work presents a systematic study on how the choice of vision encoder, training strategy, LLM backbone, and training data influences performance on fine-grained visual classification tasks—such as Flowers and ImageNet. Through a series of well-controlled experiments, the authors derive several useful insights. Leveraging these findings, they train a model that achieves strong performance on fine-grained classification benchmarks, although it still lags behind Qwen-VL, likely due to differences in training data scale.

**Strengths:**

* This paper investigates an important problem: which components of MLLMs influence performance on fine-grained visual classification. This is a valuable and underexplored topic in the existing literature.

* The work provides several useful insights into the fine-grained visual classification capabilities of modern MLLMs, which could guide future research in designing more effective multimodal models.

* Although the paper includes numerous experiments, figures, and conclusions, they are well-organized and clearly presented.
The paper is well-written and easy to follow.

**Weaknesses:**

* Although the authors investigate several factors influencing MLLM performance on fine-grained visual recognition benchmarks, the impact of data scale remains unexplored. For instance, how do different proportions of the LLaVA or Molmo datasets affect the final performance? Including such experiments would make the analysis more comprehensive.

* The conclusions drawn in this work may be valuable to the research community. However, given that commercial models are typically trained on trillions of tokens, the practical applicability of these findings might be limited, potentially reducing their instructiveness for large-scale deployments.

* The study evaluates the impact of the vision encoder by comparing CLIP and DFN, but does not include comparisons with other recently proposed and powerful encoders—such as SigLIP, AIMv2, InternViT, or SAIL-ViT. Extending the analysis to these models could further strengthen the paper's insights.

**Questions:**

In Figure 6, what do the different gray points represent?

---

### Official Review · Reviewer_koPe · 2025-11-01

**Soundness:** 2
**Presentation:** 3
**Contribution:** 2
**Rating:** 2
**Confidence:** 3

**Summary:**

This paper dives into the critical area of fine-grained perception for VLMs. This capability is essential for real-world applications where data is long-tailed and requires precise classification. Although traditional CV models are well-evaluated on these tasks, recent Large Multimodal Models often overlook this important capability. To fill this gap, the authors conducted extensive evaluations of 15 VLMs on four fine-grained classification benchmarks. Furthermore, they performed detailed ablation studies to clearly identify the key components and training strategies that dominate VLMs' fine-grained performance, leading to several important experimental findings.

**Strengths:**

* The paper is well-written and easy to follow.
* The paper is well-motivated, as the fine-grained perception capabilities is actually very important for a large multimodal models.
* Both the evaluations and ablation studies are extenisve and solid.

**Weaknesses:**

* Despite the comprehensive experimentation, the paper lacks novel insights or contributions, no new benchmarks or novel methods were proposed. The work simply reuses existing benchmarks, reformulates them, and evaluates existing models.
* LMMs are developing very rapidly, but the VLMs evaluated in the paper are outdated (e.g., LLaVA-1.5, Qwen2VL), and the insights provided may not be applicable to current VLMs.
* The experimental findings do not bring new insights, and are similar to most common knowledge about LMMs.

**Questions:**

Please see the limitations above.

---

### Meta-Review · Area_Chair_d9mn · 2026-01-10

**Summary:**

This paper evaluates numerous recent VLMs on fine-grained classification benchmarks, identifies factors causing the disconnect between fine-grained knowledge and other vision benchmarks, and finds that better vision encoders and pretraining data disproportionately improve fine-grained performance. It aims to fill the gap that VLMs trail in traditional image classification benchmarks.

**Reviewer Concerns:**

Addressed Concerns: None

Outstanding Concerns: 1. Lack of novel insights/contributions; 2. Outdated VLMs evaluated; 3. Incomplete dataset/encoder comparisons; 4. Insufficient technical depth; 5. Undermeasured prompt sensitivity

**Reviewer Scores:**

The reviewers' scores will maintain since the authors didn't provide a rebuttal.

---

### Decision · Program_Chairs · 2026-01-26

Reject